# Detecting Variants in the NBN Gene While Testing for Hereditary Breast Cancer: What to Do Next?

**DOI:** 10.3390/ijms22115832

**Published:** 2021-05-29

**Authors:** Roberta Zuntini, Elena Bonora, Laura Maria Pradella, Laura Benedetta Amato, Michele Vidone, Sara De Fanti, Irene Catucci, Laura Cortesi, Veronica Medici, Simona Ferrari, Giuseppe Gasparre, Paolo Peterlongo, Marco Sazzini, Daniela Turchetti

**Affiliations:** 1Center for Studies on Hereditary Cancer, Department of Medical and Surgical Sciences, University of Bologna, 40138 Bologna, Italy; roberta.zuntini2@unibo.it (R.Z.); elena.bonora6@unibo.it (E.B.); l.pradella@temaricerca.com (L.M.P.); amato.l@virgilio.it (L.B.A.); michele.vidone@gmail.com (M.V.); simo.ferrari@unibo.it (S.F.); giuseppe.gasparre3@unibo.it (G.G.); 2Medical Genetics Unit, IRCCS Azienda Ospedaliero—Universitaria di Bologna, 40138 Bologna, Italy; 3Department of Biological, Geological and Environmental Sciences, University of Bologna, 40126 Bologna, Italy; sara.defanti2@unibo.it (S.D.F.); marco.sazzini2@unibo.it (M.S.); 4Interdepartmental Centre Alma Mater Research Institute on Global Challenges and Climate Change, University of Bologna, 40126 Bologna, Italy; 5IFOM, Fondazione Istituto FIRC di Oncologia Molecolare, 20139 Milan, Italy; irenecatucci@hotmail.it (I.C.); paolo.peterlongo@ifom.eu (P.P.); 6Department of Oncology and Hematology, Azienda Ospedaliero Universitaria di Modena, 41125 Modena, Italy; cortesi.laura@policlinico.mo.it (L.C.); veronica.medici@gmail.com (V.M.)

**Keywords:** hereditary breast cancer, NBN, nibrin, variants

## Abstract

The NBN gene has been included in breast cancer (BC) multigene panels based on early studies suggesting an increased BC risk for carriers, though not confirmed by recent research. To evaluate the impact of NBN analysis, we assessed the results of NBN sequencing in 116 BRCA-negative BC patients and reviewed the literature. Three patients (2.6%) carried potentially relevant variants: two, apparently unrelated, carried the frameshift variant c.156_157delTT and another one the c.628G>T variant. The latter was subsequently found in 4/1390 (0.3%) BC cases and 8/1580 (0.5%) controls in an independent sample, which, together with in silico predictions, provided evidence against its pathogenicity. Conversely, the rare c.156_157delTT variant was absent in the case-control set; moreover, a 50% reduction of NBN expression was demonstrated in one carrier. However, in one family it failed to co-segregate with BC, while the other carrier was found to harbor also a probably pathogenic TP53 variant that may explain her phenotype. Therefore, the c.156_157delTT, although functionally deleterious, was not supported as a cancer-predisposing defect. Pathogenic/likely pathogenic NBN variants were detected by multigene panels in 31/12314 (0.25%) patients included in 15 studies. The risk of misinterpretation of such findings is substantial and supports the exclusion of NBN from multigene panels.

## 1. Introduction

After the discovery of BRCA1 and BRCA2 as the genes responsible for most inherited breast cancer (BC) cases, international research efforts have failed to identify additional high-risk genes that could explain BC clustering in non-BRCA1/2 families. However, many moderate- or low-risk genes have been proposed, most of which belong to the same DNA repair pathway as BRCA1 and BRCA2. With the advent of next generation sequencing, the opportunity to simultaneously analyze multiple genes, with progressively declining costs, has led to the introduction of, as a routine DNA test for hereditary BC, multigene panels including genes with low or unknown penetrance. This, however, raises concerns about the interpretation and clinical use of variants in those genes.

Among the genes commonly included in breast cancer panels is NBN, which encodes the protein nibrin: one of the components of the MRN complex, which is essential for DNA double-strand break repair. Indeed, since the identification of NBN as the causative gene of a severe recessive condition, the Nijmegen Breakage Syndrome [1], heterozygous carriers have been reported to be at increased risk of several cancer types, including BC [2]. Known NBN mutations were found in 2 of 97 BCs among French Canadian families with no BRCA1/2 mutations [3], and association studies have provided evidence of an association between specific SNPs and haplotypes of NBN and the risk of BC in US white non-Hispanic BC patients under age 56 and in Taiwanese women affected by BC [4,5]. However, most evidence is provided by studies on recurrent NBN variants: founder NBN mutations were detected more frequently in Polish and Finnish BC cases than in healthy controls [6,7,8], and the deletion c.657_661delACAAA, relatively common among populations of Slavic descent, has been associated with a relative risk for BC of 2.7 [9]. In addition, since the majority of studies have been carried out in Eastern European populations, their results may not be generalizable to populations with different genetic backgrounds, as supported by recent evidence that in the Polish population the risk conferred by the c.657_661delACAAA is strongly affected by the genotype of another founder NBN variant: E185Q [10].

For the Italian population, to the best of our knowledge, there is no evidence available on the association between NBN variants and BC.

We therefore aimed to evaluate the potential impact of a wide-spread use of NBN analysis for the genetic diagnosis of Italian BC patients by: (1) assessing the frequency and exploring the pathogenic role of NBN variants in a sample of Italian patients with familial and/or early-onset BC; (2) reviewing the literature on NBN variants detected through multigene panel test use.

## 2. Results

### 2.1. NBN Sequence Variants 

A total of 116 patients negative for mutations in the BRCA1 and BRCA2 genes were tested for NBN variants. The frequency of common variants in our patients was not higher when compared to public databases (Appendix A).Three patients (2.6%) carried rarer variants that were of unknown significance at the time of their detection: one patient (Patient 1) carried the missense variant c.628G>T, while two apparently unrelated patients (Patients 2 and 3) harbored the same frameshift c.156_157delTT variant.

### 2.2. Clinical and Family History of Carriers of NBN Variants 

Patient 1 was diagnosed at the age of 29 years with an invasive ductal carcinoma of her left breast (stage pT2N0M0; G2, ER 90%, PgR 1%). Her family history was relevant, with her mother having died from BC that was diagnosed when she was 30 years old (Figure 1A); the maternal grandmother was reported to have died from abdominal cancer that was diagnosed at the age of 40 years. She was disease-free after a follow-up of 10 years.

Patient 2 was a member of a large BC family; she underwent right mastectomy for BC at the age of 40 (pathology data not available) and was disease-free after 40 years. Two paternal aunts and two cousins had developed BC at peri- or post-menopausal ages (Figure 2). Her two daughters tested negative for the NBN variant, as well as an affected cousin, and, consequently, her affected daughter, thus excluding the co-segregation of the variant with BC in the family.

Patient 3 developed an invasive ductal BC (pT2N2M0; G3, ER 85%, PgR 50%, MIB-1 65%, HER2 3+) at the age of 31 years and died from disease progression at age 46. Both the parents had died of cancer: the mother was reported as having gastric cancer, while the father had esophageal cancer. In the paternal line, one cousin died from BC in the fifth decade of life (Figure 3A). A search for the 266delTT mutation in the sister, who was cancer-free at 57 years, revealed that she did not carry the mutation. An analysis of the 266delTT variant in BC tissue failed to show loss of heterozygosity (Figure 3B).

### 2.3. In Silico Assessment of the Variants

The missense variant c.628G>T (rs61754796) is reported with an allele frequency of 0.0005 in ExAC. It results in the aminoacidic change p.Val210Phe. At the time of its detection, this variant was of unknown significance (VUS), therefore it was subjected to in silico prediction of pathogenicity using available tools, which collectively supported its neutrality.

Conversely, the frameshift variant c.156_157delTT (rs767454740) detected in Patients 2 and 3 and predicted to introduce a premature stop-codon: p.(Ser53Cysfs*9) is classified as pathogenic in ClinVar and is reported at a frequency of 0.000008 in ExAc.

### 2.4. Genotyping of the Independent Set of Cases and Controls

The frequency of the c.628G>T and c.156_157delTT NBN variants was assessed in a larger set of BC cases and controls of Italian origin. We identified 4/1390 carriers of the c.628G>T variant among cases and 8/1580 among controls, with a frequency of 0.3% and 0.5%, respectively (*p* = 0.33), which further supports its neutrality. Conversely, no other carriers of the c.156_157delTT variant were identified among 1393 cases and 1582 controls (*p* = 0.0014).

### 2.5. Expression of Variant Nibrin

NBN mRNA expression analysis was carried out using two pairs of primers: the first, between Exons 1 and 2, was located upstream to the variants, while the other one was positioned between Exons 11 and 12, downstream to the variant c.156_157delTT. This strategy was aimed at verifying whether the mRNA containing the premature translation-termination codon (PTC) induced by the frameshift was actually degraded through nonsense-mediated decay (NMD). The transcripts of Patient 1 and Patient 2 carrying c.628G>T and c.156_157delTT, respectively, one heterozygous patient carrying the Slavic variant c.657del5, and a wild-type control were analyzed.

Patient 2 showed a fifty percent reduction in NBN transcript, which is consistent with the NMD pathway having selectively degraded the allele harboring the frameshift variant. The patient with c.628G>T showed the same relative gene expression as the control. The patient heterozygous for c.657del5 showed a slight reduction in the mRNA transcript.

In order to evaluate whether protein expression was comparable to mRNA expression, we performed Western blot and relative quantification thought densitometric analysis (Appendix A). As expected, Patient 2 showed a fifty percent reduction of protein expression, similar tomRNA, while the missense variant in Patient 1 did not affect NBN protein expression. Finally, the patient heterozygous for c.657del5 showed a marked reduction of the full-length protein expression despite the slight reduction of the mRNA transcript (Figure 4).

### 2.6. Double-Strand Break Repair Assessment

In order to evaluate the effect of the c.628G>T variant on double-strand break repair machinery, we investigated DNA foci induced by γ-irradiation, using Ser139 phosphorylation of H2AX histone. The B-LCLs of Patient 1, a patient heterozygous for the Slavic mutation, and a NBN wild-type control were analyzed. As expected, the patient with the Slavic mutation showed an increased number of foci post-treatment when compared to the control; despite the predicted neutrality of the c.628G>T variant, Patient 1 showed a foci pattern similar to the c.657del5 carrier (Figure 1B).

### 2.7. NBN Haplotype Patterns

NBN haplotype patterns were defined to investigate whether specific allelic combinations are associated with increased BC risk and whether the two patients carrying the rare c.156_157delTT variant were related. Linkage disequilibrium (LD) analysis was conducted on the control 1000 Genomes Project Tuscan population from Italy (TSI) by considering the chromosomal interval encompassing the 13 NBN variants identified via the direct sequencing of patients. This pointed out a 12 kb LD block including five variants, whose genotypes were retrieved from the entire 1000 Genomes Project panel of worldwide populations and from the population under study (i.e., data from NBN sequencing and whole exome sequencing (WES)) leading to the reconstruction of 22 haplotypes (Appendix A). Three of them (i.e., h1, h7, h22) overall accounted for 65% of the patients’ chromosomes. Because their frequencies are comparable to those observed in all the examined human groups, their association with BC is not supported. However, six haplotypes collectively showing a frequency of 11% turned out to be specific to the patients. Among them, h2 and h4 carried the c.156_157delTT and diverged only for one variant. According to the inferred relationships among haplotypes (Appendix A), h4 was supposed to have originated from h3, which showed a frequency of 4% in BC patients and was observed in a single TSI chromosome. Conversely, h2 was originated by recombination, which brought the c.156_157delTT on the most common cosmopolitan haplotype (h1). Indeed, estimates of the time to the most recent common ancestor (TMRCA) inferred for these allelic combinations was approximately 160 years, suggesting that independent occurrence of the indel on similar haplotypes in such a reduced timeframe was highly improbable. These findings are in line with the shared ancestry inferred for Patients 2 and 3. In fact, despite bio-demographic data for these subjects indicating that they are unrelated by at least three generations, cryptic genetic relatedness among them was confirmed. Exome-wide proportion of shared alleles (i.e., identity by state (IBS)) was calculated and used to compute an identity by descent (IBD) kinship coefficient of 0.154. By considering that a threshold of 0.125 generally reflects a third-degree relationship and that the obtained value plausibly represents an overestimate because it was calculated on exome-wide instead of genome-wide data, we can hypothesize that Patients 2 and 3 were related some few generations ago. Coupled with the extremely low frequency of c.156_157delTT, this suggests that it arose very recently and was locally transmitted to different, but closely related, haplotype backgrounds due to recombination.

### 2.8. Whole Exome Sequencing Analysis

In order to investigate the presence of variants in other genes potentially responsible for BC predisposition, WES was performed in the three patients carrying NBN variants. Mean coverages were of 42.11×, 44.29×, and 61.7×, respectively, ensuring that >97% of the targeted bases were covered at least 5×. A filter was set to select non-synonymous, splice-site SNVs or indels in cancer-predisposing genes (listed in Appendix A) with minor allele frequency <1% in the 1000Genomes (www.1000genomes.org), Exome Variant Server (EVS; https://evs.gs.washington.edu/), and Exome Aggregation Consortium databases (accessed on April 2020) or in our institutional exomes collection (approximately 500 individuals).

In Patient 1, WES failed to detect other variants in addition to the known NBN c.628G>T.

Conversely, Patient 2 was found to also carry the BRIP1 missense variant c.139C>G (p.Pro47Ala) (rs28903098; MAF = 0.00024 in ExAc). This variant, predicted as “damaging” by both PolyPhen-2 and SIFT, is reported as a Class 3 VUS (ClinVar). Segregation analysis of this variant showed that it is not shared by the affected cousin (III-2 in the pedigree), while it has been transmitted to one of the two daughters (both in their 60s and asymptomatic): IV-12 (Figure 2).

Notably, Patient 3 carried the splice-site c.1101-2A>T variant in TP53. This variant, which is not present in the unaffected sister, was neither reported in public databases, except as a somatic mutation in COSMIC, nor in our in-house database. However, a different substitution of the same nucleotide (c.1101-2A>G; rs587781664) is classified as pathogenic (ClinVar). Moreover, she carried a missense variant c.359T>C (p.Met120Thr) in the RAD51B gene (rs142567687; MAF = 0.0003 in ExAc); as no evidence exists on the pathogenicity of this variant, it is regarded as a VUS; however, it is predicted as “benign” by PolyPhen-2 and “tolerated” by SIFT.

### 2.9. Review of the Literature

Sixteen papers, published between November 2014 and April 2020, reported variants identified in the NBN gene through NGS-based multigene panel testing of BC patients. Their results are summarized in Table 1. Out of 12314 patients included in 15 studies providing detailed descriptions of the NBN variants detected, 31 (0.25%) carried NBN variants classified or predicted as probably pathogenic/pathogenic (C4/C5) [11,12,13,14,15,16,17,18,19,20,21,22,23,24,25,26]. This is consistent with Kurian et al. [23] who report a rate of 0.35% (95% CI 0.21–0.55) for pathogenic NBN variants among 5436 patients, whereas NBN VUS were detected in 2.8% (95% CI 2.4–3.3). Among carriers of pathogenic variants, 14 (45%) carried the recurrent variant c.657_661delACAAA, 2 the c.127C>T variant, while the other 15 had unique variants, most of which are extremely rare in the general population. Indeed, five were not reported in population databases, nine had frequencies of 0.005% or lower, while the other one is a common variant in the Ashkenazi population and was found in a patient of that ancestry. As shown in Table 1, carriers of variants had very variable features (by age of onset, family history, etc.), with high heterogeneity even among carriers of the Slavic variant.

## 3. Discussion

The introduction of NBN gene analysis in clinical genetic tests for suspected hereditary BC has been mainly based on evidence obtained on definite variants in specific populations; in the attempt to assess the expected impact of detecting NBN variants through multigene panels in Italian patients, we analyzed the NBN sequence in a series of BC patients who previously tested negative for BRCA1/2 and reviewed the literature on multigene panel testing including NBN.

Unlike other studies [4,5], we failed to find any significant difference in NBN SNP frequency and haplotypes distribution in comparison to controls and/or the general population.

However, we found NBN variants with potential clinical implications, according to the knowledge to the date of their detection, in 3 out of 116 BC patients. The first patient carried the variant c.628G>T, which was reported as probably pathogenic in databases such as LOVD. We investigated the significance of this variant first through in silico predictions, which supported the neutrality of the variant, then by genotyping a large set of cases and controls, where the variant was detected more frequently in controls than in cases. Furthermore, expression and co-localization studies of the variant protein showed no alterations. Collectively, our findings supported the neutrality of c.628G>T variant, consistent with evidence provided by later studies, which led to the current classification of c.628G>T as a likely benign variant (https://databases.lovd.nl/shared/variants/0000611721#00014315, accessed on May 2021). Similarly, the recurrent I171V variant, formerly associated with a significant increase in BC risk, was eventually suggested to be neutral [27]. This highlights a substantial risk of misinterpretation of variants for which initial evidence supports an association with cancer risk, until the actual meaning is conclusively established by evidence.

Despite the neutrality of the NBN variant, DNA repair in our patient was suggested to be impaired by the study of foci after irradiation; furthermore, her clinical and family history of early-onset cancers was highly suspicious of an inherited predisposition to cancer. However, WES did not detect any genetic variant explaining those features, which leads to the hypothesis that she may carry a defect in the non-coding part of her genome or a copy-number variant undetected by sequencing approaches and underlines the incomplete sensitivity of current diagnostic tools.

The other two patients carried the same variant c.156_157delTT. While the effect of this variant in impairing protein synthesis has been confirmed by expression studies, its role in cancer development was not supported by our findings. Indeed, for Patient 2, belonging to a large BC kindred, the variant did not co-segregate with BC in the family branch we were able to investigate; equally, Li et al. detected in a BC patient a frameshift NBN variant, which, did not co-segregate with BC in the family [16]. The other patient carrying the c.156_157delTT (Patient 3) was found to also carry a likely pathogenic variant in TP53, which is a plausible cause for her early-onset, fatal breast cancer, and, possibly, for other cancers that occurred in her family. Since both these patients, although apparently unrelated, carried the same variant, we first investigated the frequency in our independent case-control set, without finding any additional carrier and thus supporting it as a rare variant in the population of Northern Italy, to which the patients belong. However, IBD gave results consistent with genetic relatedness, suggesting that a common ancestor explains the sharing of this rare variant by the two patients.

Of note, WES in Patient 2 also showed a variant of unknown significance in BRIP1, implying that testing this patient using a wide multigene panel would have provided two findings of difficult interpretation.

Summarizing, in our series of BC patients, testing NBN failed to provide any clinically meaningful result, while creating concern and uncertainties and requiring further investigations.

Coherently, several position statements do not comprise NBN among the genes to be included on cancer panel tests [28,29].

As an ultimate confirmation, two large case-control studies have recently excluded an association of NBN variants, including the c.657_661del, with increased BC risk [30,31]

Based on the latter evidence, and given the troubles raised by the detection of NBN variants, as shown in the experience here described, we strongly believe that NBN should be removed from multigene cancer panels, which, according to the papers we reviewed, may lead to detect NBN variants classified as pathogenic in a proportion as high as 1 in 400 patients.

In general, the NBN story should be a warning in regard of the risk of introducing new cancer gene tests into clinical practice without consolidated evidence of clinical utility, as previously emphasized by Easton and colleagues [32].

## 4. Patients and Methods

### 4.1. Patient Recruitment

A consecutive series of patients with familial and/or early-onset breast cancer referred to the Cancer Genetics Clinic of the Hospital S.Orsola-Malpighi, Bologna and who had undergone complete BRCA1/2 analysis (sequencing and MLPA) in Modena and Bologna (two hubs of the Emilia-Romagna regional network [33]) with negative results were selected for the molecular analysis of NBN. 

### 4.2. NBN Sequence Analysis

PCR amplification of all the exons and exon–intron boundaries of the NBN gene was performed under standard conditions (FastStartTaq DNA Polymerase, Sigma-Aldrich, Saint Louis, MI, USA). The primer sequences are listed in Appendix A. The PCR products were sequenced using a Big Dye Terminator v1.1 Cycle Sequencing Kit (Thermo Fisher Scientific, Waltham, MA, USA) and run on an ABI3730 Genetic Analyzer (Thermo Fisher Scientific, Waltham, MA, USA).The sequence NM_002485 was used as a reference.

### 4.3. In Silico Variant Assessment

In order to predict the functional impact of missense variants, the ExPASy portal (https://www.expasy.org/) was used as a reference point to access the bioinformatic tools needed for the implementation of the Chou–Fasman and the GOR algorithms. Freely available tools, namely PolyPhen2 (http://genetics.bwh.harvard.edu/pph2/), PROVEAN (http://provean.jcvi.org/index.php), and MutPred (http://mutpred.mutdb.org/), were also used (Alla accessed November 2016).

### 4.4. Genotyping of Independent Sets of Cases and Controls

The frequency of the variants identified in the population under study was assessed in two independent sets: (i) female individuals affected with BC as the first diagnosed malignancy, negative for pathogenic mutations in BRCA1 and BRCA2, with family history of the disease and/or early-onset diagnosis, and recruited at the Medical Genetics Unit of the Fondazione IRCCS Istituto Nazionale dei Tumori (INT) and the Division of Cancer Prevention and Genetics of the Istituto Europeo di Oncologia in Milan; (ii) female consecutive blood donors used as controls, recruited at the Unit of Immunohematology and Transfusion Medicine of INT and the Associazione Volontari Italiani Sangue in Milan. Genotyping was performed at the Cogentech qPCR Facility in IFOM, Fondazione Istituto FIRC di Oncologia Molecolare, using two custom TaqMan SNP Genotyping Assays (TechnThermo Fisher Scientific, Waltham, MA, USA). Specifically, reactions were performed in a total volume of 8 uL, containing 20 ng of genomic DNA, 4 uL of GTX Press Mastermix 2×, and 0.2 uL of TaqMan SNP assay 40×. Real-time PCR was carried out on the 7500 Sequence Detector System (TechnThermo Fisher Scientific, Waltham, MA, USA) as follows: pre-PCR step of 20 s at 95 °C, 40 cycles of 3 s at 95 °C and 30 s at 60 °C. For each 96-well plate, three duplicate samples, one single non-DNA blank control, and one single positive control sample were included. Statistical analysis of frequencies was performed using Fisher’s exact test, with differences being considered as significant for *p* values < 0.05.

### 4.5. Gene Expression Analyses

#### 4.5.1. Real-Time PCR

Total RNA was isolated from peripheral blood lymphocytes using RNeasy Mini Kit (Qiagen, GmbH, Hilden, Germany). A total of 500 ng RNA was reverse transcribed into cDNA using random hexamer primers and a Transcriptor First Strand cDNA Synthesis kit (Roche Diagnostics, Indianapolis, IN, USA). Quantitative real-time PCR was performed using ABI PRISM 7500 (Thermo Fisher Scientific, Waltham, MA, USA) and Fast SYBR Green Master Mix (Thermo Fisher Scientific, Waltham, MA, USA), according to the manufacturer’s instructions. For frameshift variants, two distinct primer pairs were designed to analyze one cDNA segment upstream and another one downstream to the specific DNA change. The primer sequences are listed in Appendix A. For quantification of *NBN* expression, *ACTB* was used as an endogenous control. Relative quantification was performed using the comparative method. The results are expressed using the ΔΔCt method.

#### 4.5.2. Western Blotting

Total proteins were extracted from EBV-immortalized B cell lines (B-LCLs). Cells (10^6^) were suspended in lysis buffer (Tris-amino-metano base (TBS) 1× pH 7.6, Triton X-100 1% (Merck KGaA, Darmstadt, GermanyFluka), Proteinase Inhibitor Cocktail (Roche Diagnostics, Indianapolis, IN, USA)). Cell lysates were precleared by centrifugation at 16,000× *g* for 30 min. Protein concentration was determined using the Bio-Rad Protein Assay reagent (Bio-Rad Laboratories Inc., Milan, Italy). Proteins from total lysates were separated by 10% SDS–PAGE and transferred onto polyvinylidenefluoride membranes (PVDF) (GE Healthcare Life Sciences, Milan, Italy). Western blotting analysis was performed with antibodies against ACTB (1:2000) (Merck KGaA, Darmstadt, GermanyFluka) and NBN (1:2000) (GeneTex Irvine, CA, USA), according to the manufacturer’s instructions. The chemiluminescence signals were acquired with ChemiDoc™ XRS+ molecular imaging apparatus (Bio-Rad Laboratories Inc., Milan, Italy), and relative quantification of protein concentration was performed using densitometric analysis; data were analyzed by ImageJ software.

#### 4.5.3. Double-Strand Break Repair Assessment

Confocal laser scanning microscope analysis was performed on B-LCLs. Cells were exposed to a radiation dose of 2 Gy using a ^137^Cs source (CIS bioindustries, IBL 437C (89–294). Unirradiated control and 2 Gy irradiated cells were incubated for 1 h at 37 °C and 5% CO_2._

The cells were fixed in 4% paraformaldehyde/PBS for 30 min, permeabilized in 0.5% Triton X-100/PBS for 15 min and blocked in blocking buffer (1% bovine serum albumin in PBS) for 30 min. Staining was performed using an anti-γH_2_AX antibody (1:500, Merck KGaA, Darmstadt, Germany) for 2 h at 4 °C. After three 10 min washes, the cells were incubated with the secondary antibody (Alexa Fluor 546 goat anti-mouse IgG diluted 1:500, in 1% BSA; Thermo Fisher Scientific, Waltham, MA, USA). The distribution of γ-H_2_AX foci was visualized using confocal laser scanning microscopy (Zeiss LSM 510 Meta).

#### 4.5.4. Haplotype Reconstruction and Evolutionary Analyses

All the variants detected in the population under study were considered to perform LD analysis. Genotype data for these variants were retrieved for 107 healthy Italian individuals (TSI) sequenced by the 1000 Genomes Project [34] and used as a reliable proxy of a control population showing comparable ancestry with respect to the examined patients. Pairwise r^2^ LD values were computed for each SNP pair and blocks of high LD (i.e., mean r^2^ > 0.8) were identified with the Haploview package [35]. The five variants included in the observed high LD block were then used to statistically infer haplotypes by means of the Bayesian algorithm implemented in PHASE v2.1.1 [36].

Relationships among haplotypes of carriers of NBN variants and those observed in the control population were explored by drawing a median joining network of haplotypes using the Network software (http://www.fluxus-engineering.com) and by inferring the ancestral haplotype according to the chimpanzee allelic state at the examined loci. The mutation rate per site per year (µ = 9.017 × 10^−10^) of the genomic region including the considered variants was calculated by dividing total nucleotide divergence (Dxy = 0.0108) between 1000 Genomes Project worldwide human samples and the chimpanzee sequence by twice the divergence time between the species (i.e., roughly six million years). The computed mutation rate implied the occurrence of one mutation every 180,923 years and was used to calculate the rho statistic [37] according to the formula: 1/(µ × length of the considered genomic region in bp). This enabled approximate TMRCA estimates for the haplotypes of interest to be obtained.

#### 4.5.5. Whole Exome Sequencing

WES was performed in the three patients carrying rare NBN variants: 100 ng of genomic DNA extracted from peripheral blood was fragmented according to the Nextera^®^ Rapid Capture Enrichment protocol (Illumina Inc., San Diego, CA, USA). The fragmentation was verified using the Agilent Bioanalyzer and exomes were captured by hybridization using the coding exome exon kit (Nextera). A total of 500 ng of libraries were sequenced with the Illumina HiScan SQ platform at 100bp paired-ends. After that, reads were aligned with BWA to the reference genome (hg19). Aligned reads were treated for realignment and base quality score recalibration with GATK and for duplicate removal with PicardTools (http://picartools.sourceforge.net). The alignment statistics were collected with SAMtools and GATK. Coverage statistics over the targeted regions were calculated with GATK. Variant calling and filtering by quality was performed by GATK. Variants passing quality filters were annotated separately against NCBI RefGene (http://www.ncbi.nlm.nih.gov) and UCSC KnownGene (http://genome.ucsc.edu).

#### 4.5.6. Review of the Literature

A search was performed in PubMed (National Center for Biotechnology Information, US National Library of Medicine) to identify reports on NGS-based analysis of the NBN gene by using the following search equation: (“NBN” OR “NBS1”) AND (“breast cancer”). The final search was performed on 6 November 2020 and produced 211 references. Among those, papers reporting results of NBN germline analysis through NGS-based multigene panels were selected and data on prevalence and type of variants detected were extracted together with the clinical data of carriers, if available. The population frequencies and pathogenicity classification of reported variants were then assessed by checking public databases (gnomAD, ClinVar, LOVD).

## Figures and Tables

**Figure 1 ijms-22-05832-f001:**
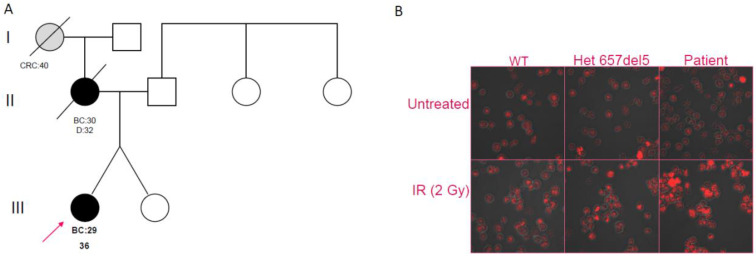
(**A**) Pedigree of Patient 1, showing early-onset breast cancer in the proband and her mother and early-onset colorectal cancer in the maternal grandmother (BC = Breast Cancer; CRC = Colorectal Cancer). (**B**) Confocal analysis on B-LCLs. Untreated cells of a wild-type individual (WT), a c.657del5 heterozygote, and Patient 1 are shown in the upper panes; in the lower panes, respective cells are shown after irradiation treatment (IR).

**Figure 2 ijms-22-05832-f002:**
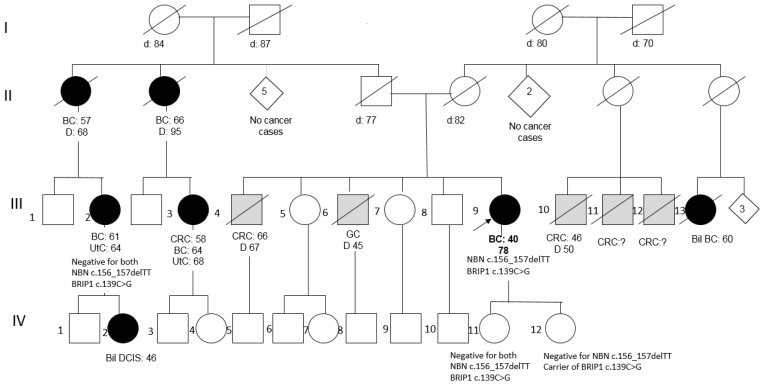
Pedigree of Patient 2 (BC = Breast Cancer; Bil = Bilateral; DCIS = Ductal Carcinoma In Situ; CRC = Colorectal Cancer; GC = Gastric Cancer; UtC = Uterine Cancer).

**Figure 3 ijms-22-05832-f003:**
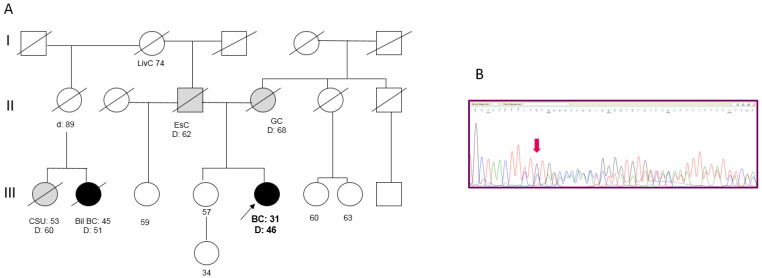
(**A**) Pedigree of Patient 3. (BC = Breast Cancer; Bil = Bilateral; EsC = Esophageal Cancer; GC = Gastric Cancer; LivC = Liver Cancer; CSU = Cancer of Site Unknown). (**B**) Targeted sanger sequencing of DNA extracted from BC tissue of Patient 3: electropherogram showing the c.156_157delTT at the heterozygous state.

**Figure 4 ijms-22-05832-f004:**
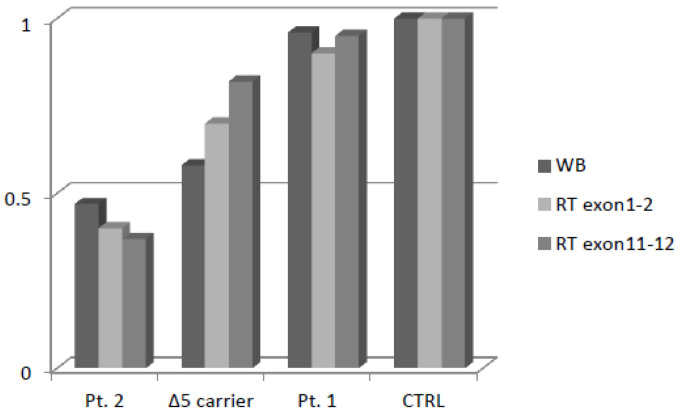
Densitometric analysis of NBN protein expression and mRNA Real-Time expression.

**Table 1 ijms-22-05832-t001:** Studies reporting variants in the NBN genes detected through multigene analysis.

Reference	Country	Study Population	Testing Type	N Patients	N with NBN Variants (%)	Specific Variants (N)	Proband Cancer History	Variant Frequency in the Study	Population Frequency (gnomAD) §	*p* Value	Current Classification (ClinVar/LOVD)
[11]	France	Pts with suspected hereditary BC	NGS panel	708	8 (1.13%)	c.1142delC (1)	ND	0.14%	0.005% ^1^	0.01	C4/C5
						c.156_157delTT (1)		0.14%	0.004% ^1,2^	0.01	C4/C5
						c.37+5G>A (3)		0.42%	0.56% ^3^	0.82	C1/C2
						c.38-10T>A (1)		0.14%	0.03% ^3^	0.50	C3
						c.657_661delACAAA (1)		0.14%	0.04% ^1,2,4,5^	0.69	C5
						c.788T>C (1)		0.14%	0.04% ^1,2,6^	0.68	C3
[12]	USA	Pts previously tested for BRCA mut	NGS panel (42 genes)	198	2 (1.01%)	c.643C>T (2)	(1) BC age 50; FH: BC, Mel (2) no cancer; FH: BC, CRC, OC	1.01%	0.49% ^3^	0.59	C3
[13]	Poland	Pts with TNBC (158) or suspected hereditary, NTNBC (44)	Screening of 36 mutations in 8 genes	202	7 (3.47%)	c.657_661delACAAA (1)	NTNBC age 63	0.49%	0.04%	0.14	C5
						c.511A>G (6 *)	4: TNBC age 44–81				
							2: NTNBC age 49, 59	2.97%	0.30% ^3^	<0.0001	C1
[14]	USA	Pts referred for BRCA and tested negative	Commercial NGS panels (25–29 genes)	1046	2 (0.19%) only deleterious variants reported	c.657_661delACAAA (1)	BC age 54, PC; FH: BC, PrC, Mel, PC, LC, others BC age 49; FH: BC	0.10%	0.04%	0.91	C5

						c.1142delC (1)		0.10%	0.005%	0.07	C4/C5


[15]	USA	Pts with early-onset BC (<40) BRCA-negative	NGS panel (22 genes)	278	1 (0.36%)	c.664T>C	BC age 37, Leu 39; FH: PrC, Mel	0.36%	0.004% ^1^	<0.0001	C3
[16]	Australia	Familial BC pts	NGS panel (19 genes)	684	1 (0.15%)	c.698_701delAACA (1)	BC age 42; FH: PrC	0.15%	0.004% ^1,6^	0.01	C5
[17]	USA	BC pts	NGS panel (25 genes)	488	1 (0.2%)	c.127C>T (1)	TNBC age 56; FH: LC, CNS, PC	0.2%	0.006% ^1,4,6^	0.01	C5
[18]	USA	BC pts with Ashkenazi ancestry	NGS panel (23 genes)	1007	1 (0.1%)	c.1903A>T (1)	BC age 42	0.1%	0.15% ^7^	0.96	C5
[19]	Germany	BC/OC pts	NGS panel (14 genes)	581	6 (1.03%)	c.1397+1delG (1)	NTNBC age 35; FH: BC NTNBC age 43; FH: BC NTNBC age 52; FH: BC (1)BC age 53; FH: BC (2) BC age 49; FH: BC (3)TNBC/OC age 56/66; FH: BC, LC, Leu	0.17%	0.0008% ^2^	<0.0001	C5
						c.2028delT(1)		0.17%	ND	-	ND (C4)
						c.2097dupT (1)		0.17%	ND	-	ND (C4)
						c.657_661delACAAA (3)		0.52%	0.04%	<0.0001	C5


						c					
[20]	China	Women with personal or familial history of BC	NGS panel (27 genes)	240	1 (0.42%)	c.2140C>T (1)	BC age 33	0.42%	0.005% ^1,7^	<0.0001	C5
[21]	Germany	BC pts tested negative for BRCA	NGS panels (8 genes selected)	5589	12 (0.21%) only truncating variants reported	c.123delC (1)	ND	0.02%	0.0008% ^1^	0.15	C5
						c.211_212insGA (1)		0.02%	0.0008% ^1^	0.15	C5
						c.657_661delACAAA (7)		0.13%	0.04%	0.008	C5
						c.1141del (1)		0.02%	ND	-	ND (C4)
						c.1396del (1)		0.02%	ND	-	ND (C4)
						c.1651dup (1)		0.02%	0.002% ^6^	0.29	C5
[22]	Germany	BC pts tested negative for BRCA	NGS panel (94 genes)	237	1 (0.4%) VUS excluded	c.657_661delACAAA	BC age 32	0.4%	0.04%	0.19	C5
[23]	US	BC/OC pts	NGS panels (11 genes selected)	5436	ND (0.35%) path ND (2.5%) VUS	ND					
[24]	Israel	BC pts tested negative for founder BRCA variants	Commercial NGS panels (30–83 genes)	144	1 (0.7%) VUS excluded	c.966C>G	ND (BRCAPRO prob: 0.6%)	0.7%	ND	-	ND (C4)
[25]	Italy	Male BC pts	NGS panel (24 genes)	81	1 (1.2%)	c.547G>A	Male BC age 49; FH: BC	1.2%	0.003% ^1,7^	<0.0001	C3
[26]	China	BC pts	Virtual panel from WES	831	1 (0.1%)	c.127C>T	ND	0.1%	0.006% ^1,4,6^	0.07	C5

Abbreviations: N: number; BC: Breast Cancer; TNBC: Triple-Negative Breast Cancer; NTNBC: Non-Triple-Negative Breast Cancer; OC: Ovarian Cancer; PeC: Peritoneal Cancer; TC: Fallopian Tube Cancer; Mel: Melanoma; CRC: Colorectal Cancer; Leu: Leukemia; LC: Lung Cancer; CNS: Central Nervous System Cancer; PC: Pancreatic Cancer; PrC: Prostatic Cancer; FH: Family History; WES: Whole Exome Sequencing; ND: Not Described. * One carried also a BRCA1 pathogenic variant; another one a CHEK2 variant. § Comparison populations: ^1^ European—Non-Finnish; ^2^ Latino; ^3^ All; ^4^ Finnish; ^5^ Other; ^6^ African; ^7^ Ashkenazi. (Statistical analysis performed by Graphpad software 5.0:chi-square test with Yate’s correction *p* value two-tailed.)

## Data Availability

Data is contained within the article or Appendix A.

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
