# Peer review of "Detecting Variants in the NBN Gene While Testing for Hereditary Breast Cancer: What to Do Next?"

_ijms, 2021, doi:10.3390/ijms22115832_

Round 1

Reviewer 1 Report

In this paper a full comprehensive analysis of NBN germline mutations was performed to conclude that the NBN gene should not be included in breast cancer multigene panels  as suggested by previous studies.

116 BRCA1/2-negative breast cancer patients were analysed and 3 were found carriers of P or LP variants in NBN .

One was carrier of a missense and the remaining 2 carried the same fs mutations.

The NBN gene has been included in Breast cancer multigene panel since several years and even if initial studies suggested an increased BC risk for carriers recent researches lead to opposite conclusion.

Even if the object of this study is not extremely new the authors carried out the investigation in an very constructive way not only presenting their results, on a cohort of 116 italian women, but also  performing a fully comprehensive revision of literature.

Minor suggestions :

Results

Figure 1: the results of in silico analysis should be removed and eventually discuss in the text. Electropherogram can be removed

In general figure 1, 2 and 3 legends need to be shortened eventually reducing the pedigree description because it is already in the main text.

Figure 4: please use better quality picture of WB or delete it

Paragraph “NBN haplotype patterns”: this paragraph needs to be shortened and the abbreviations need to be explained.

Author Response

Comment 1

Figure 1: the results of in silico analysis should be removed and eventually discuss in the text. Electropherogram can be removed

Response: Thank you for these suggestions: the figure has been amended accordingly

Comment 2

In general figure 1, 2 and 3 legends need to be shortened eventually reducing the pedigree description because it is already in the main text.

Response: Agreed: legends have been shortened

Comment 3

Figure 4: please use better quality picture of WB or delete it

Response: Unfortunately, we did not have the opportunity to obtain better quality pictures for WB. We have deleted this photo from the figure and moved it into the supplementary materials section

Comment 4

Paragraph “NBN haplotype patterns”: this paragraph needs to be shortened and the abbreviations need to be explained.

Response: The paragraph has been reorganized for clarity and abbreviations explained

Reviewer 2 Report

In the manuscript entitled “Detecting NBN variants while testing for Hereditary Breast Cancer: what to do next?” by Zuntini et al. authors explore the potential impact of Nibrin (NBN) analysis in the genetic diagnosis of breast cancer patients. Authors provide both the results on the pathogenic role of NBN variants in a cohort of 116 Italian BRCA-negative breast cancer patients and review and discuss previously published results.

General comment:

By analyzing NBN sequence in a series of breast cancer patients and reviewing the literature on NBN testing, authors convincingly support the case for the failure of NBC analysis to provide any clinically meaningful result.

Minor issues:

Title page: I suggest not to use abbreviation in the title

  • Figure 1: Define type of cells shown in panel D and type of analysis. Include size bars. Quantification analysis should be provided for each sample.
  • Line 149: “several available tools”. Please, define.
  • Figure 4: use 0.5 instead 0,5 notation
  • Improve organization and information of figure legends, including supplementary figure

Author Response

Comment 1

Title page: I suggest not to use abbreviation in the title

Response: the title has been rephrased to add clarity

Comment 2

Figure 1: Define type of cells shown in panel D and type of analysis. Include size bars. Quantification analysis should be provided for each sample.

Response: The type of cells has been specified in the legend. Unfortunately, we do not have quantitative data.

Comment 3

Line 149: “several available tools”. Please, define.

Response: we have rephrased in “it was subjected to in silico prediction of pathogenicity using available tools”. Details on the tools are reported in the Methods section

Comment 4

Figure 4: use 0.5 instead 0,5 notation

Response: The typo has been corrected.

 Comment 5

Improve organization and information of figure legends, including supplementary figure

Response: The legends in the manuscript and supplementary material have been re-organized